# Evaluation of the Psychometric Properties of the Musculoskeletal Health Questionnaire (MSK-HQ) in a Population of Kitesurfers: A Cross-Sectional Study

**DOI:** 10.3390/medicina60121995

**Published:** 2024-12-02

**Authors:** Marco Alessandro Tonti, Alessandra Carlizza, Giovanni Galeoto

**Affiliations:** 1Faculty of Medicine, UniCamillus, International Medical University in Rome, 00131 Rome, Italy; matonti2000@gmail.com (M.A.T.); alessandra.carlizza@unicamillus.org (A.C.); 2Department of Human Neurosciences, Sapienza University of Rome, 00185 Rome, Italy; 3IRCCS Neuromed, Via Atinense 18, 86077 Pozzilli, Italy

**Keywords:** musculoskeletal injury, sport injury, quality of life, psychometrics, questionnaire

## Abstract

*Background and Objectives*: Musculoskeletal disorders affect a large portion of the population worldwide. The Musculoskeletal Health Questionnaire (MSK-HQ) is a helpful tool for assessing the health state of patients with these disorders. The primary goal of this study is to evaluate the psychometric properties of the MSK-HQ-IT in a population of kitesurfers. *Materials and Methods*: The study was conducted from September 2023 to July 2024. The questionnaire was completed using an online or paper form. Data were collected by submitting both the Italian and English versions of the MSK-HQ to a global sample of kitesurfers from various countries. *Results*: A total of 102 participants were recruited, consisting of 40 professionals and 62 non-professional kitesurfers. Cervical spine discomfort was significantly more prevalent among professionals (42.5%) compared to non-professionals (24%), as well as right shoulder pain (37.5% vs. 22.5%) and right wrist pain (12.5% vs. 3.2%). Non-professionals exhibited a significantly higher prevalence of functional limitations in the lumbar spine (25.8% vs. 5%) and reported more thoracic pain (21% vs. 17.5%). These findings indicate differing biomechanical stress patterns between the two groups, with professionals showing higher upper limb strain and non-professionals experiencing more lower back issues due to harness reliance. *Conclusions*: The MSK-HQ proved to be a reliable and valid tool for assessing musculoskeletal health in kitesurfers. The study highlights distinct injury patterns between professionals and non-professionals, with professionals being more prone to upper limb injuries and non-professionals showing a higher prevalence of lumbar and thoracic spine issues. These findings emphasize the need for targeted injury prevention strategies. Further research should focus on expanding the sample size and investigating long-term impacts of repetitive high-impact landings on musculoskeletal health in kitesurfers.

## 1. Introduction

Kitesurfing, also known as kiteboarding, is a popular water sport practiced globally, combining elements of surfing, windsurfing, wakeboarding, and paragliding. This dynamic and rapidly growing sport enables surfers to perform jumps, tricks, and ride waves [1].

The unique demands of kitesurfing, such as controlling the kite while maintaining balance on the board, place participants at increased risk of sustaining specific orthopedic injuries. A typical kitesurfing session involves repetitive movements, high-intensity activity and impact landings, plus a very important exposure to varying wind and water conditions. A cohort study from 2020 analyzed 194 kitesurfers of various skill levels and riding styles and age were surveyed prospectively during a full kitesurf season. The results showed a total of 177 injuries mainly reported in the lower limbs: the foot and ankle were the most common site of injury (56 out of 177 injuries, 31.8%), followed by the knee (25 out of 177 injuries, 14%). Regarding upper limb injuries, injuries of the hands, wrists, and fingers (25 out of 177 injuries, 14%) predominated over shoulder, arm, and forearm injuries (17 out of 177 injuries, 9.7%) [2]. Most injuries are reported by different authors to occur during tricks or high jumps. Additionally, many of the injuries are caused by technical difficulties controlling the kite and board in challenging weather conditions. A comparative study conducted over a two-year period in the Netherlands highlighted a significantly higher injury rate amongst kitesurfers (7.0/1000 h) in comparison with windsurfers (5.2/1000 h) in the same environmental conditions, proving that kitesurfers are more prone to injuries than participants in other water sports regardless of the environmental factors [3].

Different studies have investigated the epidemiology of injuries in recreational kitesurfers [4]. No studies have investigated the epidemiology of injuries in professional kitesurfers and compared the differences between the professional and non-professional populations. More importantly, no studies evaluate the quality of life in individuals suffering from musculoskeletal disorders, emotional factors that are very important to consider, especially in the professional population. Numerous scales investigate the evaluation of the generic quality of life, but none are specific for evaluating the quality of life associated with musculoskeletal disorders.

The MSK-HQ is a concise questionnaire designed to enable individuals with musculoskeletal conditions (such as arthritis or back pain) to report their symptoms and quality of life in a standardized manner. It was developed collaboratively by the Arthritis Research UK Primary Care Sciences Research Centre at Keele University and the University of Oxford, with active input and feedback from individuals with arthritis and musculoskeletal conditions, as well as clinicians and academics. The MSK-HQ allows patients and their therapists to monitor overall musculoskeletal health, assess progress, and respond to treatment. Additionally, the questionnaire allows particular aspects of musculoskeletal health to be addressed, ensuring a holistic approach to patient needs, but it is also possible to consider individual components of the score, such as sleep quality or mood [5]. A correct use of the MSK-HQ may support people to report a wider range of their symptoms to their clinical team than could be measured by a simple clinical screening. The measurement properties of the MSK-HQ have been validated across various patient samples with musculoskeletal disorders [6]. This scale has also been validated in Italian (MSK-HQ-IT) on a healthy population and has shown excellent psychometric properties [7]. The MSK-HQ-IT has also been used to evaluate a population of professional basketball players with excellent outcomes [8].

Given the specific physical demands and injury risks associated with kitesurfing, it is crucial to validate the MSK-HQ for this particular group. Validating the MSK-HQ for kitesurfers will ensure that the questionnaire is both reliable and relevant, providing accurate assessments that can inform clinical decisions, guide interventions, and ultimately enhance athlete health and performance. Each sport imposes unique functional overloads on the joints most involved in its specific motor activities. Therefore, it is essential to have validated tools capable of evaluating the specific impact that joint limitations have on the quality of life of athletes. This validation process will allow for a more precise understanding of how musculoskeletal disorders affect kitesurfers, thereby facilitating targeted strategies to manage and prevent these injuries effectively.

The primary goal of this study is to evaluate the psychometric properties of the MSK_HQ-IT in a diverse population of kitesurfers. The secondary aim is to evaluate and compare the prevalence and impact of musculoskeletal disorders between professional and non-professional kitesurfers through the MSK-HQ scale and a questionnaire to investigate limitations and joint pain of the various body segments.

## 2. Materials and Methods

This cross-sectional study was conducted by the R.E.S. (Ricerca Evidenza e Sviluppo) research group from the Sapienza University of Rome (Italy); in the last few years, the R.E.S. group has been involved in carrying out systematic reviews and validating outcome measures.

### 2.1. Participants and Procedure

The study was conducted over a 10-month period, from 1st September 2023 to 31st July 2024. Participants were recruited from various kitesurfing clubs and associations globally, starting with the GKA Kitesurf World Championship held in Brazil. According to the “Consensus-based standards for the selection of health measurement instruments” (COSMIN) guidelines and consistent with previous studies of the MSK-HQ [8], a minimum sample size of 100 subjects was considered adequate for this study. All participants provided informed consent [9] before taking part in the study.

Inclusion criteria included active kitesurfers aged 18 and above and a full understanding of the respective MSK_HQ and MSK-HQ-IT scales. The participants in the professional group are professional athletes that compete in the World Tour, Continental, and National Championships, including world-renowned athletes, such as the current World Champions in the different disciplines: freestyle, wave, and park riding; and four of the multi-time Female World Champions. On the other hand, the minimum requirement for the non-professional group was the ability to ride upwind.

Both categories, professionals and non-professionals, completed either the MSK-HQ-IT or the MSK-HQ, depending on their preferred language. The administration of the questionnaire was carried out directly by physical therapists to avoid comprehension errors. All participants who filled out the English version of the questionnaire were able to understand the English language.

### 2.2. Outcome Measures

Data were gathered by submitting a questionnaire to a population of kitesurfers. The questionnaire used in this study consisted of four parts:Informed consent form: Participants were informed about the study’s objectives and methods [9].Demographic data sheet: This included personal information such as age, gender, previous traumas/orthopedic surgeries, group classification: professional/non-professional, other associated sports activities, and weekly hours of kitesurfing.A musculoskeletal disorders questionnaire to collect specific data from body regions that exhibit functional limitations and/or pain. If pain is present, participants are asked to give a score of 0 to 10, where 0 indicates no pain and 10 indicates unbearable pain. The body parts investigated are the cervical, thoracic, and lumbar regions of the spine and thoracic cage; other parts considered are the shoulders, elbows, wrists, and hands for the upper body and the hips, knees, ankles, and feet for the lower body.MSK-HQ: This part assessed musculoskeletal health, including pain and functional limitations in different body regions. Both the Italian (MSK-HQ-IT) and the original English versions were used to accommodate a diverse population. The MSK-HQ is a patient-reported outcome measure (PROM) designed to assess the overall impact of musculoskeletal disorders on individuals. It evaluates multiple domains of health affected by musculoskeletal conditions, including pain, physical function, and psychological well-being. In detail, its domains assess the following areas: Pain and discomfort: evaluates the level and frequency of pain experienced by the patient; Physical function: assesses the ability to perform daily activities and physical tasks; Stiffness: measures the degree of stiffness and its impact on movement; Work and daily activities: considers how musculoskeletal issues affect work, chores, and daily routines; Sleep: assesses the impact of musculoskeletal disorders on sleep quality; Fatigue: evaluates the level of fatigue related to musculoskeletal conditions; Emotional well-being: measures the psychological impact, including feelings of anxiety or depression related to MSK health; Social participation: assesses how musculoskeletal issues affect social interactions and activities. MSK-HQ consists of 14 items, and for each of them, the score goes from 4 to 0, where 4 means that the patient does not experience pain, 3 means little pain, 2 means moderate pain, 1 means intense pain, and 0 is high-intensity pain [7].

### 2.3. Statistical Analysis

The software used for the statistical analyses was SPSS Statistics version 27. The demographic and clinical characteristics were calculated as mean ± SD or percentage where appropriate. The analysis for the evaluation of psychometric properties was conducted following the directions of the COSMIN checklist for evaluating the methodological quality of studies on measurement properties [10].

Reliability was assessed using the test–retest reliability, administering the MSK-HQ-IT twice to a randomized sample of the whole sample by the same professional. A time interval of 14 days was considered long enough to prevent recall. The intraclass correlation coefficient (ICC) was calculated, and a value of 0.70 was considered optimal to establish consistency of responding over time. The internal consistency was examined by Cronbach’s alpha, in order to assess the interrelatedness of the items and the homogeneity of the scale. Values of Cronbach’s alpha (α) higher than 0.70 were considered acceptable as an indicator of the satisfactory homogeneity of the items within the total scale. Cross-cultural validity was evaluated using analysis of variance (ANOVA), in which the four cohorts were compared based on the total score of the MSK-HQ [7].

The floor–ceiling effect was calculated for each item of the MSK-HQ-IT, describing whether participants have scores that are at, or near, the possible lower or upper limits, respectively, preventing measurement of variance above or below a certain level. The floor effect is recognized when more than 15% of kitesurfers obtain the lowest possible score, while the ceiling effect corresponds to the achievement of the highest possible score by less than the 15% of the participants [11].

Furthermore, to assess the clinical and statistical significance of the scores obtained from the administration of the MSK-HQ-IT between individuals with musculoskeletal pain and those without, a Student’s *t*-test for independent samples was performed. A *p*-value of < 0.05 was considered indicative of a statistically significant difference between the two groups.

## 3. Results

### 3.1. Population and Demographic Characteristics

The population in the following study consists of 102 individuals. The characteristics of the sample are reported in Table 1. Only 3 out of 102 individuals were left-handed, while the remaining participants were right-handed. Therefore, the left-handed subgroup was not large enough to draw a statistically significant difference in the study.

### 3.2. Statistical Analysis: Internal Consistency and Test–Retest Reliability

The test–retest reliability was calculated on a subpopulation of 62 individuals randomized from the original population. The data showed high reliability for all items and for the total scale in Table 2.

To evaluate the internal consistency of the MSK-HQ scale, we used Cronbach’s alpha (α). The scale showed a value of 0.84, and the alpha-deleted analysis showed that all items contribute to the evaluation of the construct of the scale (Table 3).

All variables with very low prevalence (<15%) or *p*-values significantly distant from the threshold of significance (*p* = 0.05) were excluded from the detailed commentary.

### 3.3. Floor and Ceiling Effect

The floor–ceiling effect was calculated for each item of the MSK-HQ-IT; results are summarized in Table 4.

Of fourteen items, a floor effect was revealed for two out of five scores in six items and for one out of five scores in one item; for two out of four scores in four items and for one out of four scores in two items. Only item 8 achieved a ceiling effect at a score of five, while item 12 showed no effect.

### 3.4. Discriminative Power of the Test

From the statistical analysis, it was possible to deduce statistically and clinically significant differences between the two groups explored (non-professionals/professionals), as reported in Table 5.

### 3.5. Assessment of Musculoskeletal Disorders

Cervical Spine: Functional limitations in the cervical spine were reported by 16% of non-professionals and 15% of professionals, while 24% of non-professionals and 42.5% of professionals experienced cervical spine pain. This difference in pain prevalence suggests that being a non-professional may offer a protective effect against cervical pain, as indicated by the odds ratio. The *p*-value (*p* = 0.052) is close to significance, highlighting a trend toward a higher prevalence of pain in professionals.

Thoracic Spine: Pain in the thoracic spine was reported by 21% of non-professionals and 17.5% of professionals. Although the *p*-value (*p* = 0.052) did not reach formal significance, the data suggest a potential protective effect for professionals.

Lumbar Spine: The lumbar region showed the most significant difference between groups, with 25.8% of non-professionals reporting functional limitations compared to just 5% of professionals. This highlights a much higher risk for non-professionals, confirmed by the statistically significant *p*-value (*p* = 0.0007). Additionally, lumbar pain was more common among non-professionals (42%) than professionals (32.5%), further suggesting that non-professionals face greater strain in this area, even though the difference in pain was not statistically significant.

As you can see in the table below (Table 6), these findings clearly demonstrate the differences in spinal health between the two groups.

Right Shoulder: Right shoulder pain was reported by 22.5% of non-professionals and 37.5% of professionals. The odds ratio (OR = 0.60, 95% CI: 0.32–1.10) suggests that being a non-professional may provide some protection against right shoulder pain, though the *p*-value does not indicate statistical significance.Left Shoulder: Functional limitations in the left shoulder were reported by 17.7% of non-professionals and 5% of professionals. The odds ratio (OR = 3.54, 95% CI: 0.83–15.17) indicates that non-professionals may be more prone to functional limitations in the left shoulder. The *p*-value (*p* = 0.06) is close to statistical significance, highlighting a potential difference between the groups.Pain in the left shoulder was reported by 16% of non-professionals and 22.5% of professionals, but this difference was not statistically significant.Right Elbow: Right elbow pain was reported by 9.6% of non-professionals and 20% of professionals. The odds ratio (OR = 0.484, 95% CI: 0.181–1.291) suggests a potential protective effect for non-professionals, though the *p*-value does not show statistical significance.Left Elbow: 9.6% of non-professionals and 15% of professionals report left elbow pain, with no statistically significant difference.Right Wrist: Right wrist pain was reported by 3.2% of non-professionals and 12.5% of professionals. The *p*-value (*p* = 0.07) approaches statistical significance, suggesting that professionals may be more likely to experience pain in the right wrist.

As illustrated in the table below (Table 7), these results emphasize the contrasting prevalence of upper limb issues between non-professional and professional kitesurfers.

Right Knee: Functional limitations were reported by 14.5% of non-professionals and 10% of professionals, while pain was more prevalent, affecting 30.6% of non-professionals and 35% of professionals. Although the *p*-values (*p* = 0.50 and *p* = 0.65) do not show statistical significance, the high levels of pain reported in both groups suggest that the knee is a common area of discomfort, particularly among professionals.

Left Knee: For the left knee, 12.9% of non-professionals and 15% of professionals reported functional limitations. Pain was more frequent, with 22.6% of non-professionals and 32.5% of professionals affected. Although the *p*-values (*p* = 0.76 and *p* = 0.27) are not significant, the higher prevalence of pain among professionals could indicate increased strain in this group.

Right Ankle: Right ankle pain stands out, with 22.5% of professionals reporting pain compared to just 8% of non-professionals. This difference is statistically significant (*p* = 0.03), supported by an odds ratio that indicates a protective effect for non-professionals, highlighting the greater vulnerability of professionals to right ankle pain.

As shown in Table 8, these findings emphasize the notable prevalence of lower limb issues, particularly ankle pain, in professional kitesurfers.

## 4. Discussion

This cross-sectional study, classified as Level 2b of scientific evidence, aimed to validate the MSK-HQ-IT in a population of professional and non-professional kitesurfers. Level 2b studies provide moderate evidence, typically drawn from well-designed cohort or case-control studies, often conducted in multiple research settings.

While previous research has extensively focused on musculoskeletal health in athletes from other sports, there is a notable lack of studies addressing kitesurfers [1,2,3], who face unique biomechanical demands due to the nature of the sport. The most important finding of this study is to provide a valid and reliable tool useful for assessing musculoskeletal health in kitesurfers, filling an important gap in the current literature. Moreover, it is the first to assess the psychometric properties of the MSK-HQ in this specific population, representing a valuable contribution to sports medicine.

The primary objective was to validate the MSK-HQ in kitesurfers. Data from 102 participants yielded a Cronbach’s alpha of 0.84, indicating good internal consistency and reliability. This is comparable to previous validations of the MSK-HQ in different languages and populations, such as Italian (0.87) [7], which has also been validate on a population of basketball players [8] in Arabic (0.88) [12], Hungarian (0.92) [6], Turkish (0.91) [13] and Norwegian (0.86) [14] versions, which were conducted on healthy populations.

The analysis of the scoring distribution of each of the fourteen items shows a ceiling and floor effect for many of them. Most of the scores presented a floor effect (e.g., item 1 “Pain/stiffness during the day” or item 6 “Work/daily routine”), suggesting that the majority of the 102 assessed kitesurfers reported experiencing no musculoskeletal pain at the time of completing the questionnaire, indicating an overall good physical health. The same consideration can be made for the only item to present a ceiling effect (item 8 “Needing help”), in which the greatest number of the athletes interviewed found themselves forced to ask for help due to joint pain/stiffness.

As regards the cross-cultural analysis, statistically significant scores were highlighted between the group of professionals (*n* = 30) and the group of non-professionals (*n* = 62) for item 2 (Pain/stiffness during the night), item 7 (Social activities and hobbies) and item 8 (Needing help). In these specific aspects, the possible impact of a musculoskeletal disorder is worse in professional athletes than in amateurs.

The secondary objective was to investigate the prevalence of musculoskeletal disorders by assessing joint pain and functional limitations across various body regions. The cervical spine emerged as a significant concern among professional kitesurfers, with 42.5% reporting discomfort compared to 24% of non-professionals. The biomechanical demands of kitesurfing, characterized by advanced aerial maneuvers and high-speed landing impacts, place considerable stress on the cervical spine. The load-bearing capacity of the cervical muscles is crucial for stabilizing both the head and shoulders during these landings. Upon landing, the forward shift of the trunk causes the head, which weighs approximately 4.5–5 kg, to undergo rapid deceleration, increasing strain on the cervical muscles [15,16]. This is further supported by studies on axial compression and cervical spine injuries related to sports [17]. This repeated strain heightens the risk of musculoskeletal discomfort in this area.

Additionally, the asymmetry between the right and left sides in professionals is noteworthy. Professional kitesurfers place a greater load on their right upper limb during complex maneuvers, leading to a higher prevalence of right-side shoulder, elbow, and wrist pain. Right shoulder pain was reported by 37.5% of professionals, compared to 22.5% of non-professionals. This asymmetry is likely due to sport-specific motor actions that place greater strain on the right side, particularly during the flying phase of a jump. The forces transmitted through the arms contribute to strain on the neck and shoulder girdle and generate high levels of repetitive, asymmetrical loading on the right upper limb, making it more susceptible to pain and functional limitations.

Although the *p*-value for cervical pain did not reach formal significance (*p* = 0.052), the higher prevalence of cervical discomfort in professionals likely results from this repeated unilateral strain, further compounded by the rapid deceleration of the head during landings. Our data support the hypothesis that professional kitesurfers experience higher cervical pain due to the increased strain placed on their upper limbs during unhooked maneuvers. In contrast, non-professionals, who predominantly ride hooked-in, report greater thoracic and lumbar discomfort. This pattern aligns with the areas of the body where the harness distributes load, which we explore further in relation to thoracolumbar biomechanics.

Non-professional kitesurfers reported higher thoracic pain (21%) compared to professionals (17.5%), likely due to their increased reliance on the harness for support, which concentrates biomechanical loading in this region. This constant loading may predispose non-professionals to higher rates of pain and functional limitations. The lumbar spine, however, exhibited the most pronounced differences between the two groups. Non-professionals showed a significantly higher prevalence of functional limitations (25.8%) compared to professionals (5%), with a statistically significant odds ratio (OR = 5.161, 95% CI: 1.254–21.251). This suggests that non-professional kitesurfers are at a much greater risk of developing lumbar spine issues.

This biomechanical restriction concentrates the load on the lumbar spine and hips, predisposing non-professionals to functional impairments and pain. The connection between the hip and lumbar spine was further evident in this study, as non-professionals demonstrated higher levels of hip pain and functional limitations in the right hip (11.3%) compared to professionals (5%). Both the hip and lumbar spine are crucial for stabilizing the body during kitesurfing maneuvers, particularly during landings. The hip–lumbar link is consistent with findings from other sports, such as running, where restricted range of motion of hip flexion and poor hamstrings and back flexibility have been shown to significantly contribute to lower back and hip pain in athletes [2].

Non-professional kitesurfers, who may lack the physical conditioning or refined technique of their professional counterparts, are more susceptible to cumulative biomechanical stress, particularly during prolonged sessions or high-intensity maneuvers. This suggests a need for targeted prevention and rehabilitation strategies aimed at reducing the load on these areas for non-professionals [18].

A crucial difference between the two groups lies in the type of equipment used, which significantly affects the biomechanical demands on the lower limbs. Most professional kitesurfers wear boots, which firmly anchor their feet to the board, much like in snowboarding or wakeboarding. The boots offer stability, enabling athletes to land aggressively with high-speed impacts that are often explosive and forceful. These hard landings, while necessary for performing at the highest level, concentrate significant forces on the lower limbs, especially the knees and ankles. Similar injury patterns have been observed in snowboarding [19] and wakeboarding [20], where both the techniques and use of boots closely resemble those employed by professional kitesurfers. Right knee pain was reported by 30.6% of non-professionals and 35% of professionals, while left knee pain was reported by 22.6% of non-professionals and 32.5% of professionals. The right ankle emerged as a significant area of concern, with 22.5% of professional kitesurfers reporting pain compared to only 8% of non-professionals (*p* = 0.03). This statistically significant difference suggests that professional kitesurfers frequently push their physical limits, especially during competitions and intensive training. The forceful and repetitive landings inherent in high-intensity maneuvers place excessive strain on the ankle, which serves as the primary shock absorber. This predisposes professionals to overuse injuries, particularly in the lead-up to or during competitive events, where the intensity and frequency of landings are heightened.

This study has certain limitations, but it is important to highlight that a sample of 102 participants is substantial, particularly in the context of a relatively new and rapidly growing sport like kitesurfing. The inclusion of professional athletes, specifically those competing in the World Tour, the highest level of international competition, adds significant value to the study. This focus on elite male and female athletes provides a high-quality sample that is representative of the top tier of the sport, ensuring that the findings are relevant to those at the highest levels of kitesurfing performance. A potential bias of this study, though, is due to language restriction because of the administration of the questionnaire in two different languages. Future research could still benefit from expanding the sample to more diverse populations to further confirm these results and broaden the understanding of musculoskeletal health in kitesurfers. Additionally, further exploration of the biomechanical differences between amateurs and professionals and the long-term effects of repetitive high-impact landings would add depth to our understanding of injury patterns in this sport.

Addressing these limitations could enhance the validity and applicability of similar studies moving forward and offer valuable insights for refining methodologies and guiding further research in this area.

## 5. Conclusions

This study successfully validated the MSK-HQ-IT as a reliable and valid tool for assessing musculoskeletal health in kitesurfers, providing key insights into the unique biomechanical demands of the sport. The MSK-HQ-IT demonstrated strong psychometric properties.

The results highlight the importance of using standardized assessment tools across athletic populations to allow for data comparison and evidence synthesis. Additionally, this study underscores the need for targeted injury prevention strategies, particularly for professional kitesurfers who are at greater risk of upper limb and ankle injuries due to the biomechanical stresses inherent in the sport. Future research should focus on expanding the sample to more diverse populations, investigating the long-term effects of repetitive high-impact landings, and refining injury prevention and rehabilitation protocols to improve the quality of life and performance of kitesurfers. Future studies could also focus on collecting and analyzing kinematics and kinetics data to further differentiate professionals from amateurs; moreover, it would be interesting to further investigate the psychological impact of these disorders by administering specific measurement tools with which to carry out a concurrent validity analysis.

## Figures and Tables

**Table 1 medicina-60-01995-t001:** Sample characteristics.

	Test–Retest	Population
Female Gender N (%)	22 (35.5)	34 (33.3)
Professionals N (%)	29 (46.8)	40 (39.2)
Age Mean (years) ± SD	29.26 ± 9.21	31.12 ± 10.23
Height Mean (in meters) ± SD	1.75 ± 0.09	3.5 ± 17.65
Weight Mean (in Kg) ± SD	71.16 ± 12.09	70.98 ± 12.24
BMI Mean ± SD	23.01 ± 2.59	22.62 ± 3.39
Weekly physical activity (hours) ± SD	12.52 ± 6.48	12.22 ± 6.41
Daily working/study hours ± SD	6.71 ± 2.37	6.74 ± 2.36

SD: standard deviation.

**Table 2 medicina-60-01995-t002:** Test–retest analysis: Range of ICC parameters of each item for the MSK-HQ-IT.

Item	Test	Retest	ICC	CI 95%	
	Mean ± SD	Mean ± SD		Lower Bound	Upper Bound
1. Pain/stiffness during the day	2.40 ± 0.97	2.37 ± 1.00	0.966	0.945	0.980
2. Pain/stiffness during the night	3.10 ± 0.92	2.87 ± 0.82	0.883	0.813	0.928
3. Walking	3.47 ± 0.78	3.48 ± 0.78	0.987	0.978	0.992
4. Washing/Dressing	3.63 ± 0.75	3.63 ± 0.75	0.971	0.952	0.982
5. Physical activity levels	2.84 ± 0.89	2.68 ± 0.83	0.796	0.682	0.872
6. Work/daily routine	3.21 ± 0.91	3.21 ± 0.89	0.980	0.967	0.988
7. Social activities and hobbies	3.44 ± 0.78	3.44 ± 0.80	0.974	0.957	0.984
8. Needing help	3.84 ± 0.49	3.82 ± 0.50	0.967	0.945	0.980
9. Sleep	3.32 ± 0.92	3.19 ± 0.92	0.933	0.890	0.959
10. Fatigue or low energy	2.76 ± 0.95	2.56 ± 0.88	0.867	0.788	0.918
11. Emotional well-being	2.98 ± 0.93	2.97 ± 0.94	0.953	0.924	0.972
12. Understanding of your condition and any current treatment	2.69 ± 1.08	2.71 ± 1.09	0.993	0.989	0.996
13. Confidence in being able to manage your symptoms	2.58 ± 1.11	2.53 ± 1.11	0.941	0.904	0.964
14. Overall impact	2.61 ± 0.84	2.55 ± 0.90	0.959	0.933	0.975
MSK-HQ Total	42.87 ± 7.00	42.02 ± 6.92	0.985	0.975	0.991

SD: standard deviation; CI: confidence interval; ICC: intraclass correlation coefficient.

**Table 3 medicina-60-01995-t003:** Cronbach’s alpha if item deleted.

Item	Medium Scale If the Item Is Deleted	Scale Variance If the Element Is Deleted	Correct Element-to-Total Correlation	Quadratic Multiple Correlation	Cronbach’s α If the Item Is Deleted
1. Pain/stiffness during the day	41.48	39.282	0.734	0.631	0.810
2. Pain/stiffness during the night	40.85	42.840	0.489	0.533	0.828
3. Walking	40.52	43.539	0.529	0.342	0.827
4. Washing/Dressing	40.34	45.297	0.404	0.330	0.834
5. Physical activity levels	41.06	42.868	0.489	0.450	0.828
6. Work/daily routine	40.75	42.722	0.549	0.432	0.825
7. Social activities and hobbies	40.54	43.399	0.543	0.379	0.826
8. Needing help	40.23	46.553	0.367	0.309	0.836
9. Sleep	40.67	44.264	0.376	0.319	0.835
10. Fatigue or low energy	41.25	42.588	0.515	0.339	0.827
11. Emotional well-being	40.84	42.114	0.545	0.374	0.825
12. Understanding of your condition and any current treatment	41.35	43.498	0.311	0.410	0.844
13. Confidence in being able to manage your symptoms	41.58	44.286	0.249	0.437	0.849
14. Overall impact	41.29	39.992	0.728	0.660	0.812

**Table 4 medicina-60-01995-t004:** Floor–ceiling effect of the MSK-HQ-IT.

Item	Score				
	0	1	2	3	4
1. Pain/stiffness during the day	1 (1) *	15 (14.7) *	29 (28.4) *	38 (37.3) *	19 (18.6)
2. Pain/stiffness during the night		5 (4.9) *	17 (16.7)	32 (31.4)	48 (47.1)
3. Walking		2 (2.0) *	10 (9.8) *	21 (20.6)	69 (67.6)
4. Washing/Dressing		2 (2.0) *	6 (5.9) *	11 (10.8)	83 (81.4)
5. Physical activity levels		6 (5.9) *	22 (21.6)	40 (39.2)	34 (33.3)
6. Work/daily routine	1 (1.0) *	1 (1.0) *	15 (14.7)	34 (33.3)	51 (50.0)
7. Social activities and hobbies		2 (2.0) *	10 (9.8) *	23 (22.5)	67 (65.7)
8. Needing help		0 (0)	6 (5.9) *	5 (4.9) *	91 (89.2) **
9. Sleep	1 (1.0) *	1 (1.0) *	17 (16.7)	21 (20.6)	62 (60.8)
10. Fatigue or low energy	1 (1.0) *	6 (5.9) *	28 (27.5)	44 (43.1)	23 (22.5)
11. Emotional well-being	1 (1.0) *	4 (3.9) *	15 (14.7)	34 (33.3)	48 (47.1)
12. Understanding of your condition and any current treatment	2 (2.0) *	16 (15.7)	23 (22.5)	30 (29.4)	31 (30.4)
13. Confidence in being able to manage your symptoms	4 (3.9) *	18 (17.6)	28 (27.5)	29 (28.4)	23 (22.5)
14. Overall impact	1 (1.0) *	9 (8.8) *	25 (24.5)	45 (44.1)	22 (21.6)

SD: standard deviation; CI: confidence interval; ICC: intraclass correlation coefficient. * floor effect. ** ceiling effect.

**Table 5 medicina-60-01995-t005:** Cross-cultural validity: mean score (standard deviation)—Student *t*-test for independent samples.

Item	NP (*n* = 62)	P (*n* = 40)	
	Mean (SD)		*p* Value
1. Pain/stiffness during the day	2.81 (1.00)	2.23 (0.86)	0.448
2. Pain/stiffness during the night	3.34 (0.94)	3.00 (0.78)	0.026 *
3. Walking	3.60 (0.76)	3.45 (0.75)	0.390
4. Washing/Dressing	3.79 (0.66)	3.60 (0.67)	0.055
5. Physical activity levels	3.13 (0.82)	2.80 (0.96)	0.139
6. Work/daily routine	3.40 (0.74)	3.15 (0.95)	0.462
7. Social activities and hobbies	3.63 (0.58)	3.35 (0.95)	0.001 *
8. Needing help	3.92 (0.33)	3.70 (0.68)	0.001 *
9. Sleep	3.50 (0.97)	3.23 (0.88)	0.103
10. Fatigue or low energy	2.90 (0.88)	2.65 (0.89)	0.483
11. Emotional well-being	3.47 (0.74)	2.83 (1.01)	0.127
12. Understanding of your condition and any current treatment	2.68 (1.07)	2.75 (1.21)	0.202
13. Confidence in being able to manage your symptoms	2.47 (1.14)	2.50 (1.15)	0.790
14. Overall impact	3.03 (0.83)	2.35 (0.92)	0.118

NP: Non-professionals; P: professionals; SD: standard deviation, * *p* < 0.05.

**Table 6 medicina-60-01995-t006:** Assessment of musculoskeletal disorders.

Spinal Region	NP/P	Functional Limitation						Pain					
		NO (n)	YES (n)	Total	OR: CI 95%	X^2^	*p*	NO (n)	YES (n)	Total	OR: CI 95%	X^2^	*p*
Cervical	NP	52	10	62	1.075 [0.424–2.727]	0.023	0.878	47	15	62	0.569 [0.322–1.006]	3.785	0.052
	P	34	6	40				23	17	40			
Thoracic	NP	47	15	62	0.569 [0.322–1.006]	3.785	0.052	49	13	62	1.198 [0.523–2.743]	3.785	0.052
	P	23	17	40				33	7	40			
Lumbar	NP	46	16	62	5.161 [1.254–21.251]	7.243	0.0007 *	36	26	62	1.290 [0.756–2.201]	0.917	0.338
	P	38	2	40				27	13	40			
Rib Cage	NP	60	2	62	0.323 [0.062–1.680]	2.015	0.156	54	8	62	1.290 [0.416–4.004]	0.197	0.657
	P	36	4	40				36	4	40			

NP: Non-professionals; P: professionals, * *p* < 0.05; OR: odds ratio; CI: confidence interval.

**Table 7 medicina-60-01995-t007:** Assessment of musculoskeletal disorders.

Superior Limb	NP/P	Functional Limitation						Pain					
		NO (n)	YES (n)	Total	OR: IC 95%	X^2^	*p*	NO (n)	YES (n)	Total	OR: IC 95%	X^2^	*p*
Shoulder R	NP	53	9	62	1.16 [0.41–3.21]	0.08	0.77	48	14	62	0.60 [0.32–1.10]	2.66	0.13
	P	35	5	40				25	15	40			
Shoulder L	NP	51	11	62	3.54 [0.83–15.17]	3.55	0.06	52	10	62	0.71 [0.32–0.61]	0.65	0.42
	P	38	2	40				31	9	40			
Elbow R	NP	59	3	62	1.93 [0.21–17.96]	0.35	0.55	56	6	62	0.484 [0.181–1.291]	2.18	0.14
	P	39	1	40				32	8	40			
Elbow L	NP	58	4	62	2.58 [0.30–22.26]	0.81	0.37	56	6	62	0.645 [0.224–1.861]	0.66	0.41
	P	39	1	40				34	6	40			
Wrist R	NP	61	1	62	0.98 [0.95–1.02]	0.65	0.42	60	2	62	0.258 [0.053–1.267]	3.27	0.07
	P	40	0	40				35	5	40			
Wrist L	NP	58	4	62	0.93 [0.88–0.99]	2.68	0.10	56	6	62	1.290 [0.342–4.867]	0.1	0.70
	P	40	0	40				37	3	40			
Hand R	NP	60	2	62	0.968 [0.92–1.01]	1.31	0.25	60	2	62	0.645 [0.095–4.397]	0.20	0.65
	P	40	0	40				38	2	40			
Hand L	NP	59	3	62	1.93 [0.21–17.96]	0.35	0.55	58	4	62	1.290 [0.248–6.720]	0.09	0.76
	P	39	1	40				38	2	40			

NP: Non-professionals; P: professional, *p* < 0.05.

**Table 8 medicina-60-01995-t008:** Assessment of musculoskeletal disorders.

Inferior Limb	NP/P	Functional Limitation						Pain					
		NO (n)	YES (n)	Total	OR: IC 95%	X^2^	*p*	NO (n)	YES (n)	Total	OR: IC 95%	X^2^	*p*
Hip R	NP	55	7	62	2.258 [0.494–10.328]	1.19	0.27	55	7	62	0.903 [0.308–2.651]	0.03	0.85
	P	38	2	40				35	5	40			
Hip L	NP	57	5	62	1.075 [0.272–4.253]	0.01 *	0.92	56	6	62	0.774 [0.253–2.369]	0.20	0.65
	P	37	3	40				35	5	40			
Knee R	NP	53	9	62	1.452 [0.479–4.399]	0.44	0.50	43	19	62	0.876 [0.498–1.540]	0.21	0.65
	P	36	4	40				26	14	40			
Knee L	NP	54	8	62	0.860 [0.323–2.294]	0.09	0.76	48	14	62	0.695 [0.366–1.320]	1.23	0.27
	P	34	6	40				27	13	40			
Ankle R	NP	57	5	62	0.461 [0.157–1.352]	2.08	0.15	57	5	62	0.358 [0.129–0.992]	4.28	0.039 *
	P	33	7	40				31	9	40			
Ankle L	NP	57	5	62	1.613 [0.329–7.917]	0.36	0.55	59	3	62	0.645 [0.137–3.040]	0.31	0.56
	P	38	2	40				37	3	40			
Foot R	NP	62	0	62	1.026 [0.976–1.078]	1.56	0.21	57	5	62	1.613 [0.329–7.917]	0.36	0.55
	P	39	1	40				38	2	40			
Foot L	NP	61	1	62	0.984 [0.953–1.016]	0.65	0.42	61	1	62	0.645 [0.042–10.023]	0.10	0.75
	P	40	0	40				39	1	40			

NP: Non-professionals; P: professionals, * *p* < 0.05.

## Data Availability

Data supporting this study’s findings are available from the corresponding author upon reasonable request.

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
