# Peer review of "Evaluation of the Psychometric Properties of the Musculoskeletal Health Questionnaire (MSK-HQ) in a Population of Kitesurfers: A Cross-Sectional Study"

_medicina, 2024, doi:10.3390/medicina60121995_

Round 1
Reviewer 1 Report
Comments and Suggestions for Authors
I think it is a very interesting work with relevant conclusions. I consider it appropriate to make some minimal changes related to:
1) Why if (lines 93 and 94)? : “The primary goal of the study was to evaluate MSK-HQ and MSK-HQ-IT…” in the conclusion only MSK-HQ-IT is cited
2) Since the work is based on the MSK-HQ form, during the introduction it would be convenient to briefly explain what the items of this form are or at least say that there are 14 (which would be beneficial when understanding the successive tables). Likewise, understanding that the MSK-HQ and MSK-HQ-IT are basically the same form with small variations due to translation, since both are cited, it should be explained whether or not there are differences between the two; especially because in the objectives it talks about both forms separately (I insist that later in the conclusions, it only cites MSK-HQ-IT).
3) The general rules for scientific works indicate that, both in tables and figures, the titles of the tables must contain the information necessary to correctly interpret the table or figure without resorting to the text. This fundamentally allows non-expert readers to better interpret what the table summarizes and in my opinion would improve the quality of the work. I had a hard time understanding several of the tables without thoroughly reviewing the text.
4) Line 55 states “Different studies…” but there are no citations. Next (line 60) it states: “Numerous scales…” which also has no citation.
Author Response
See the document attached below.

Reviewer 2 Report
Comments and Suggestions for Authors
Dear all,
Thank you for the opportunity to review this manuscript. The topic aligns well with the objectives of Medicina and provides valuable insights for researchers and professionals in the fields of Sports Medicine and Sports Traumatology. However, I believe some information could benefit from further clarification. I have outlined some points listed below:
Introduction
While the introduction provides interesting context, I believe it could benefit from further refinement to align more closely with the study’s objectives and focus. Specifically, it might be useful to frame the introduction around the core research question and key aspects of your work, which will help readers understand how the study contributes to existing knowledge.
Lines 35-44 need to be supported by references.
Lines 47-49 the reported results need references to confirm.
Same in lines 55-62, lines 63-77, and lines 83-92.
The whole abstract, of 2 pages of background, you used only 5 references!. Using references is essential to providing a foundation of existing knowledge. References enhance credibility by providing verifiable sources that readers can consult to better understand the research background and support for the paper's claims and to avoid plagiarism by giving credit to original authors.
Materials and Methods
Please, could you specify and mention the study design (e.g., randomized controlled trial, cross-sectional, etc.), I prefer that you describe.
Line 108: ‘a minimum sample size of 100 subjects’. The authors didn’t mention how power analysis was determined or if the sample size was appropriate to distinguish significant effects.
As you used the Italian (MSK-HQ-IT) and the original English versions to accommodate a diverse population, then you should exclude participants who were unable to understand or read Italian or English and add to the exclusion criteria, or could the authors clarify more details on the limitations and potential biases (e.g., language restrictions).
Results
Line175: ‘3.2. Statistical analysis’ this title needs to be checked.
I recommend rearrange the structure of this section into:
3.1. Demographic characteristics,
3.2 Internal consistency and test-retest reliability,
3.3 Assessment of muscoloskeletal disorders,
Adding:
3.4 Floor and ceiling effect,
3.5 discriminative power of the questionnaire.
And 3.6 Construct validity
Discussion
Thank you for starting the discussion with the study's aim. Could you please follow this statement with a brief sentence summarizing the key findings, such as, 'The most important finding of this study was...'
The researcher outlined several limitations of this study that future researchers should consider. Addressing these limitations could enhance the validity and applicability of similar studies moving forward and offer valuable insights for refining methodologies and guiding further research in this area.
References
The authors used only 25 references. Few references may reflect lower quality; however, exceptions exist, such as in emerging fields with limited research or in brief reports where fewer references may still add value. I suggest aiming for a balanced approach—moderation in using references, picking references that best serve the research ideas.
References are not adequate [(e.g., Title of the article should be Abbreviated Journal Name (italic) Year (bold)] to Medicina which recommends the ACS style guide for references; please follow all inserted references.
Best wishes,
Comments on the Quality of English Language
NA
Author Response
Please see the document attached below.

Reviewer 3 Report
Comments and Suggestions for Authors
Dear Author, please recheck and answer the below items:
In the introduction part:
1. How does the current study address the unique physical demands of kitesurfing compared to other water sports in terms of injury risks and specific musculoskeletal outcomes?
2. Given that previous research has focused mainly on recreational kitesurfers, what specific distinctions in injury patterns or quality of life issues are anticipated between recreational and professional kitesurfers in this study?
3. How does the MSK-HQ differ from other commonly used musculoskeletal health assessments in capturing the psychological and physical impact of injuries in kitesurfers?
4. What specific psychometric properties of the MSK-HQ are being evaluated in this study, and how will these contribute to its validation for a kitesurfing population?
In the Materials and Methods Parts:
1. How did the study ensure a representative sample of both professional and non-professional kitesurfers, especially given the diversity in skill levels, disciplines, and geographical locations?
2. Were there any additional measures taken to assess comprehension of the MSK-HQ and MSK-HQ-IT for non-native speakers of English or Italian, beyond direct administration by physical therapists?
3. How were potential confounding factors, such as prior injuries or the number of weekly hours spent kitesurfing, accounted for in the analysis of musculoskeletal health outcomes?
4. What specific procedures were implemented to ensure consistency in pain scoring across participants, given the subjective nature of pain and functional limitations as assessed in the Musculoskeletal Disorders Questionnaire and MSK-HQ?
In the Discussion part:
1. Sample Size and Representativeness: Given the sample of 102 participants, which is a modest size, how generalizable are the findings to the broader kitesurfing population, especially for non-elite, recreational participants? Would a larger sample size or a more diverse participant pool potentially affect the findings?
2. Biomechanical Considerations: The study discusses specific biomechanical demands and injury patterns for professional versus non-professional kitesurfers. Could more detailed kinematic or kinetic data have been included to support these conclusions, or might future studies incorporate such data to clarify these biomechanical differences?
3. Longitudinal Impact: The discussion mentions the potential long-term effects of repetitive high-impact landings. Could a follow-up study be designed to track these effects over time, possibly through a longitudinal study on injury incidence and musculoskeletal health degradation in kitesurfers?
4. Psychosocial Impact: While the study highlights physical injuries and functional limitations, the MSK-HQ also assesses psychological well-being. Could the authors expand on whether there were any notable differences in emotional well-being between professionals and non-professionals, given their varying physical demands and levels of musculoskeletal discomfort?
Author Response
See the document attached below.

Reviewer 4 Report
Comments and Suggestions for Authors
Dear Authors,
it is my pleasure to review your study.
Article titled "Evaluation of the Psychometric Properties of the Musculoskeletal Health Questionnaire (MSK-HQ) in a Population of Kitesurfers: A Cross-Sectional Study" raises an interesting topic but I have a lot of doubts.
1.First of all, the article should be prepared in accordance with the journal's guidelines: the reference format should be corrected.
2.The abstract should contain the purpose of the study. It should be improved.
3.The introduction requires correction. Every statement should be supported by a citation.
4.On line 42: please provide the reference of this study at the beginning of the paragraph, not at the end. It makes it easier to read and find a specific reference. This applies to the entire text.
5.The discussion of the MSK-HQ questionnaire should be in section 2. Materials and Methods, not in the introduction.
6.The introduction lacks basic information about typical injuries among Kitesurfers. It needs to be corrected.
7.No indication to cite publications 5-11 (line: 101-102)? What is the relevance of these references to this manuscript?
8.On line 104: "The study was conducted from September 2023 to July 2024." - please specify the duration of the study more precisely.
9.In table no. 1-6 numerical values ​​should be presented after a dot and not after a comma. It requires correction.
10.Please provide the name of the ethics committee that approved the study. This information should be in section 2. and at the end of the manuscript in the section "Institutional Review Board Statement:".
11.Did the participants complete the survey on their own? Did someone help them? Was there someone who could explain things they didn't understand? The patient may not understand some medical issues, anatomical names, etc. This can significantly affect the results of the presented study and raises some controversy.
12.In table no. 1 what does "Test-retest" mean? It is not clear.
13.In Table 2 the abbreviation ICC should be explained.
14.In table 2 and 3 what does "Item" mean? It is not clear.
15.In Table 4 it is not clear what the values ​​"Yes" and "No" refer to, whether it is a number "n", a percentage, etc. It should be corrected.
16.On line 282, what does "(1,2,16)" mean?
17.On lines 283-285: "This study, therefore, fills a crucial gap in literature by providing one of the first in-depth assessments of musculoskeletal health in kitesurfers." I cannot agree with this statement. The health status of kitesurfers is not determined by a doctor, so this assessment is not in-depth. It should be corrected.
18.On lines 294-302: there is a missing reference in this paragraph. It should be supplemented.
The same situation is in the following paragraphs. Missing references should be filled in. This is essential to improve the scientific quality of the manuscript.
19.The conclusion is extensive and unreadable. It should be corrected.
In my opinion, the manuscript contains many shortcomings and requires significant corrections.
Author Response
See the document attached below. Thank you

Round 2
Reviewer 4 Report
Comments and Suggestions for Authors
Dear Authors,
thank you for the changes you made.
The quality of the manuscript is higher, but the manuscript requires further revision.
1.The references in the introduction still need to be carefully proofread. Reference 1 is cited, then 3. Reference 2 is missing in the text.
2.in lines 44-48: "The results showed a total of 44 of 177 injuries mainly reported in the lower limbs: the foot and ankle were the most 45 common site of injury (56, 31.8%), followed by the knee (25, 14.1%). Regarding upper limb 46 injuries, injuries of the hands, wrists and fingers (25, 14%) predominated over shoulder, 47 arm, and forearm injuries (17, 9.7%)[3]." is there information in brackets, what does the first value mean, what does the second value mean in percentage? Furthermore, we have in one case (25, 14.1%) and in the other (25, 14%). This should be corrected, unified.
3.The use of a questionnaire in a study is a part of the assessment method, so it is usually included in section 2. Materials and Methods. Measurement methods can be mentioned in the introduction, but a detailed explanation should be provided in the second section.
4.Line 110: "The study was conducted over a 10-month period, from September 2023 to July 2024." the date is still general, please specify from which September to which July.
5.What does the symbol "*" mean in table 4?
Author Response
|
Reviewer Comment |
Response |
Line # |
|
Reviewer #1 |
||
|
The references in the introduction still need to be carefully proofread. Reference 1 is cited, then 3. Reference 2 is missing in the text. |
We apologize for the oversight. The missing citation (Reference 2) has now been correctly added to the introduction.
|
|
|
In lines 44-48: "The results showed a total of 44 of 177 injuries mainly reported in the lower limbs: the foot and ankle were the most 45 common site of injury (56, 31.8%), followed by the knee (25, 14.1%). Regarding upper limb 46 injuries, injuries of the hands, wrists and fingers (25, 14%) predominated over shoulder, 47 arm, and forearm injuries (17, 9.7%)[3]." is there information in brackets, what does the first value mean, what does the second value mean in percentage? Furthermore, we have in one case (25, 14.1%) and in the other (25, 14%). This should be corrected, unified. |
Modifications have been made to the text, as follows: “(56 out of 177 injuries, 31.8%)”
And “(25 out of 177 injuries, 14%)”
And “(25 out of 177 injuries, 14%)”
And “(17 out of 177 injuries, 9.7%)” |
Line 46
Lines 46-47
Lines 47-48
Lines 48-49 |
|
The use of a questionnaire in a study is a part of the assessment method, so it is usually included in section 2. Materials and Methods. Measurement methods can be mentioned in the introduction, but a detailed explanation should be provided in the second section. |
The manuscript has been revised accordingly, and the detailed explanation of the questionnaire has been moved to Section 2, Materials and Methods, as recommended. |
|
|
Line 110: "The study was conducted over a 10-month period, from September 2023 to July 2024." the date is still general, please specify from which September to which July. |
The manuscript has been revised as suggested.
“from 1st September 2023 to 31st July 2024” |
\Lines 102-103 |
|
What does the symbol "*" mean in table 4? |
The table has been modified as suggested. “* floor effect ** ceiling effect” |
Lines 212-213 |
Please see the file attached below.
